# Metabolic Changes Associated with Different Levels of Energy Deficits in Mediterranean Buffaloes during the Early Lactation Stage: Type and Role of the Main Lipid Fractions Involved

**DOI:** 10.3390/ani13142333

**Published:** 2023-07-17

**Authors:** Anastasia Lisuzzo, Maria Chiara Alterisio, Elisa Mazzotta, Paolo Ciaramella, Jacopo Guccione, Matteo Gianesella, Tamara Badon, Enrico Fiore

**Affiliations:** 1Department of Animal Medicine, Production and Health, University of Padua, 35020 Legnaro, Italy; anastasia.lisuzzo@phd.unipd.it (A.L.); matteo.gianesella@unipd.it (M.G.); bad.tamy@gmail.com (T.B.); 2Department of Veterinary Medicine and Animal Productions, University of Napoli “Federico II”, 80137 Napoli, Italy; mariachiara.alterisio@unina.it (M.C.A.); paolo.ciaramella@unina.it (P.C.); 3Istituto Zooprofilattico Sperimentale delle Venezie (IZSVE), 35020 Legnaro, Italy; emazzotta@izsvenezie.it

**Keywords:** lipid classes, negative energy balance, Mediterranean buffaloes, TLC-GC, ketosis

## Abstract

**Simple Summary:**

The mobilization of lipids from adipose tissue increases fatty acids and ketone bodies levels. The β-hydroxybutyrate is the main ketone body used to diagnose ketosis, a metabolic disorder of the transition period, in ruminants. Nevertheless, a specific cut-off for the ketosis of β-hydroxybutyrate in buffaloes and the plasma lipid fractions related to ketone bodies have not been established. The relative concentrations of not only total plasma lipids but also lipid fractions such as phospholipids, free fatty acids, triglycerides, and cholesterol esters are influenced by the mobilization of lipids. Each of these fractions has a different role in animal metabolism, influencing energy redistribution and cell metabolism and function. The present study reveals the relationship between lipid fractions and changes in metabolism and inflammation that is related to variations in lipid classes according to different levels of energy deficits in the early lactation of Mediterranean buffaloes. Furthermore, buffaloes defined as at risk of ketosis showed similarities, with ketotic cows suggesting the necessity of further investigations in these ruminants.

**Abstract:**

Cell function and energy redistribution are influenced by lipid classes (phospholipids (PLs), free fatty acids (FFAs), triglycerides (TGs), and cholesterol esters (CEs)). The aim of this study was to investigate metabolic alterations that are related to changes in lipid classes according to different levels of energy deficits in early lactating Mediterranean buffaloes (MBs). Sixty-three MBs were enrolled at the beginning of lactation using an observational study with a cross-sectional experimental design. Serum β-hydroxybutyrate (BHB) levels were used to group the animals into a healthy group (Group H; *n* = 38; BHB < 0.70 mmol/L) and hyperketonemia risk group (Group K; *n* = 25; BHB ≥ 0.70 mmol/L). Statistical analysis was performed using a linear model that included the effect of the group and body condition score to assess differences in fatty acid (FA) concentrations. A total of 40 plasma FAs were assessed in each lipid class. Among the FAs, eight PLs, seven FFAs, four TGs, and four CEs increased according to BHB levels, while three FFAs, three TGs, and one CE decreased. The changes among lipid class profiles suggested the influence of inflammatory response, liver metabolism, and the state of body lipid reserves. In addition, the possible similarities of buffaloes at risk of hyperketonemia with ketotic cows suggest the necessity of further investigations in these ruminants.

## 1. Introduction

Increased interest in buffalo-derived products led to a gradual intensification of breeding conditions, resulting in a stressful environment that is thus far unknown to this species [1]. Generally, buffaloes are more similar to beef cattle than dairy cows from a morphological and metabolic point of view [1]. Anyway, the raised intensive management systems that maximize milk and meat, genetic selection, and higher nutritional requirements led to a higher incidence of several pathologies associated with metabolic imbalance [2]. A potentially critical period is the transition period characterized by major physiological, nutritional, metabolic, and immunological changes [3,4].

Amino acids, glycerol, and fatty acids (FA) are mobilized from skeletal muscle and adipose tissue during this period [5]. Consequently, non-esterified fatty acids (NEFAs) are released into the bloodstream and are used as an energy source by peripheral tissues and organs [2]. Altered liver functions or excessive lipomobilization promotes ketone body production, especially in β-hydroxybutyrate (BHB). Serum or plasma NEFA and BHB are accepted biomarkers of excessive negative energy balance in dairy cows and ewes [6,7]. Nevertheless, a specific BHB threshold for identifying metabolic disorders such as ketosis in buffaloes has not been established [4]. Both NEFA and BHB can negatively influence inflammatory response and immune system functions, promoting an increase in other metabolic and reproductive disorders [2,4]. The mobilization of lipids from adipose tissue induces not only the relative concentrations of total plasma lipids but also lipid fractions such as phospholipids (PLs), free fatty acids (FFAs), triglycerides (TGs), and cholesterol esters (CEs) [8,9]. Each of these fractions has a different function in animal metabolism [10].

To the best of the authors’ knowledge, the blood plasmatic lipid fractions related to ketone bodies have not been investigated in buffaloes. The aim of this study was to investigate metabolic alterations that are related to changes in plasma lipid classes according to different levels of energy deficits in early lactating Mediterranean buffaloes (MBs) using gas chromatography associated with thin-layer chromatography (TLC-GC).

## 2. Materials and Methods

### 2.1. Animals and Farm

For the study’s purposes, n° 63 MBs (*Bubalus bubalis*) were selected from a single dairy farm located in Caserta (Campania Region, Italy) and characterized by an artificially induced seasonal calving herd (late winter–springtime). The sample’s size was calculated according to Friedman (1982) [11]: Assuming an effect size of 0.40, a correlation analysis was carried out with a 0.90 power level and a two-tailed significant level of 0.05. All animals within 50 days in milk (DIM) were randomly selected from January to April 2019, within the entire group of fresh buffaloes. All buffaloes were milked twice a day.

The selected farm was casually extracted from a group of 10 farms that regularly require consultancy services at the Veterinary Teaching Hospital—Didactic Mobile Clinic Service of the Department of Veterinary Medicine and Animal Production of Naples (Italy). The inclusion criteria for selecting the farm were as follows: (i) a similar herd size (~400 buffaloes, consistent along the year); (ii) a total mixed ration (TMR)-based feeding system, given two times/day; (iii) the absence of a regular monitoring program for metabolic diseases; (iv) housing and overall management system respecting the minimum welfare standard for buffaloes [12].

The selected barn was characterized by solid grooved concrete floors in the walking and feeding alleys. The lying area comprised elevated cubicles covered with rubber mattresses (for milking MBs) or a roofed deep straw yard area (for dry MB). The TMR included the following ingredients: dry hay, ryegrass silage (plastic-wrapped baled) and corn silage, buffalo cake (Stick-Florido^®^, Fusco Industry, Caserta District, IT—characterized by 23% of crude proteins—originating mainly from legumes and cereals; crude fat at 5.5%; crude fiber at 7.5%; ash at 6.9%; and sodium at 0.32%), and sodium bicarbonate (0.8% of the diet dry matter). The composition of TMR was determined using a portable analyzer based on near-infrared reflectance spectroscopy (AgriNIRTM Analyzer, Dinamica Generale^®^ s.p.a., Mantova, Italy).

### 2.2. Clinical Procedures and Experimental Design

The MBs selected were individually confined to a trimming chute and submitted to a complete clinical examination to exclude coexisting diseases (e.g., metritis, gastrointestinal disease, foot disorders, etc.), as performed in a previous study [13]. The four areas, including ribs, spine, hips, and tail base, where animals usually store adipose tissue were used to estimate the body condition score using a 9-point scoring system [14].

At the end of the clinical examination, blood samples were collected by carrying out jugular venipuncture with a 10 mL syringe (Becton Dickinson Hypodermic Syringes, Franklin Lakes, US—equipped with a 21-gauge needle). Some drops of blood (obtained directly by the syringe) were immediately used for a BHB test in the field (FreeStyle^®^, Abbott, Maidenhead, Berkshire, UK), while the remaining amount of blood was placed in one tube containing EDTA (Vacutainer^®^, Becton and Dickinson, Franklin Lakes, NJ, USA) and one tube containing a clot activator (Vacutainer^®^, Becton and Dickinson, Franklin Lakes, NJ, USA). Samples were then centrifuged (908× *g* × 15 min, centrifuge model DMO412, GIORGIO-BORMAC s.r.l., Carpi, IT) to obtain plasma and serum in the field. Therefore, a cross-sectional experimental design was used. 

The plasma and serum samples were immediately placed in a cool box (4 °C) and brought at the same temperature to the reference laboratory of the University of Naples within 1 h of collection for further investigations. For each plasma sample, 250 µL of plasma was immediately transferred to Eppendorf tubes (maximum capacity 1 mL/tube) containing 5 mg of pyrogallol to reduce FA oxidation [15]. The samples were stirred until the pyrogallol was completely dissolved. Also, serum samples were transferred to Eppendorf tubes (1 mL of serum/tube) without pyrogallol. Immediately after, the two aliquots (plasma and serum) were sent on dry ice to the Department of Animal Medicine, Production, and Health (MAPS) at the University of Padua (Italy), arriving within 24 h and stored at −20 °C until biochemical and TLC-GC analyses. 

### 2.3. Biochemical Analysis and Group Division

A multi-parametric analyzer was used to perform biochemical analysis (BT3500 Biotecnica Instruments S.p.a., Rome, Italy). The quantification of BHB (RANBUT RX Monza test; Randox, Crumlin, UK), NEFA (NEFA RX Monza test; Randox, Crumlin, UK), and Glucose (Glucose Monoreagent, LR; Gesan S.r.l, Campobello di Mazara, Italy) was added to the normal biochemical profile of the analyzer.

Considering that there is no specific BHB cut-off for buffaloes to identify animals as healthy and hyperketonemic, we selected a subjective cut-off, as described in the study of Fiore et al. [4]. Specifically, we used the BHB concentration reported in a previous study on healthy buffaloes at the same lactation period [16] using the mean BHB value of 0.4 mmol/L plus three standard deviations of 0.1 each to establish a subjective BHB cut-off of 0.7 mmol/L. According to the serum BHB concentrations obtained in the laboratory, MBs were then divided into the following groups: a healthy group (Group H) comprising 38 MBs with a level of BHB < 0.70 mmol/L and a group at risk of hyperketonemia (Group K) consisting of 25 MBs with BHB ≥ 0.70 mmol/L.

### 2.4. Thin Layer Chromatography Associated with Gas Chromatography (TLC-GC)

The procedure for plasma TLC-GC was similar to Fiore et al. (2020) [8] with respect to the plasma of dairy cows. Briefly, an internal standard for each lipid class (C15 for PL and C17 for FFA, TG, and CE) was added to each sample. Next, lipid extraction was performed via biphasic separation using chloroform and methanol (2:1, *v*/*v*). This solution allowed the distinction between a supernatant containing methanol and water and a subnatant containing chloroform and lipids divided by a thin protein state. To intensify the separation of the two portions, NaCl was added. Afterward, lipids were isolated using a heating block at 37 °C under a nitrogen flow to reduce fatty acid oxidation. Regarding TLC, the samples were dissolved in chloroform (100 µL) with BHT added (50 mg/L; antioxidant). The solutions were laid out on the deposition line of the TLC, which allowed lipid class separation via a different degree of affinity. The four lipid classes (PL, FFA, TG, and CE) were obtained by scraping the silica gel containing each lipid class for each sample. The lipid classes were then methylated using 3N methanolic hydrochloric acid and placed in an oven for one hour at 100 °C. Immediately after, samples were neutralized using a solution containing 10% of potassium carbonate (K_2_CO_3_), and the extracted lipids were dissolved in hexane with 50 mg/dl of BHT added. For each lipid class and sample, a total of 40 FA were identified and quantified in the splitless mode by GC using a TRACE GC/MS (Thermo Quest, Milan, Italy) equipped with a flame ionization detector (FID) and a polar fused-silica capillary column (Capillary Column Omegawax, 30 m × 0.25 mm × 0.2 µm film). Helium was used as the carrier gas at a flow rate of 1 mL/min. Data for plasma FA were calculated in mg/dL.

### 2.5. Statistical Analysis

Statistical analysis with one-way ANOVA was first used to compare the animal data and biochemical analyses between groups after evaluating the normal distribution using the Shapiro–Wilk test. The software used was R ver. 4.2.1 (R core team, Vienna, Austria).

Animal data (BCS, parity, DIM, and daily production) and biochemical data (NEFA, cholesterol (CHO), triglycerides (TGRs), glucose (GLU), γ-glutamyl-transferase (GGT), aspartate transaminase (AST), and alanine transaminase (ALT)) were included individually or associated as covariates in statistical models to assess differences in the FA profiles. These models were evaluated using the Akaike information criterion (AIC) and the Bayesian information criterion (BIC) to assess model performance. The linear model that included the fixed effect of the group and BCS as the covariate showed a lower AIC and BIC and was selected to assess the difference in FA concentrations between groups in each lipid class. A post hoc pairwise comparison among LSMEANS was performed using Bonferroni correction.

A *p*-value ≤ 0.05 was used to consider statistically significant differences, while a *p*-value between 0.05 and 0.1 was used to highlight differences with trend to significance.

The Boruta algorithm is a random forest classification algorithm that provides a numerical estimate of feature importance. During this analysis, multiple unbiased weak classifiers (decision trees) are used to perform a classification independently between decision tree parts. The importance measure of classifiers is obtained as the loss of accuracy of classification. Then, the average and standard deviation of the accuracy loss are computed [17]. This decision algorithm (R2 software ver. 4.2.3, Santa Monica, CA, USA) was used to select the FA within each lipid class that may provide a better classification for the group with BHB ≥ 0.70 mmo/L (Group K—at risk of hyperketonemia).

The consequently selected FA in each lipid class was used to obtain receiver operating characteristic (ROC) (MedCalc Sofware Ltd., Ostend, Belgium) curves to evaluate their diagnostic power and to establish the threshold value in order to identify animals at risk of hyperketonemia. The diagnostic power was assessed via the area under the curve (AUC), which identifies FA as an excellent marker when the AUC is 0.9 to 1.0; good if AUC is 0.8 to 0.9; moderate if AUC is 0.7 to 0.8; poor if AUC is 0.6 to 0.7; and fail if AUC is 0.5 to 0.6 [18]. The AUC is associated with the 95% confidence interval (CI) and the sensitivity (Se) and specificity (Sp) of the test.

## 3. Results

The animals’ data included BCS, parity, DIM, and daily production, which did not show differences between groups, except for BCS which was slightly greater in Group K compared to Group H (5.08 in Group K vs. 4.58 in Group H; *p*-value = 0.058). Regarding the biochemical analysis, BHB and AST were the only two parameters with greater concentrations in Group K (0.74 mmol/L and 164.0 u/L in Group K vs. 0.47 mmol/L and 140.0 u/L in Group H, respectively; *p*-values < 0.001 and 0.017) (Table 1). 

The differences in the FA profile of PL are shown in Table 2. All significant FAs of this lipid class (C8:0; C16:2 ω 4; C18:1 ω 9; C18:1 ω 7; C20:1 ω 11; C20:1 ω 9; C22:6 ω 3; and C24:1 ω 9) showed an increase in Group K compared to Group H. The FFA profile showed an increase of 7 FAs (C16:4 ω 1; C20:3 ω 3; C20:5 ω 3; C22:2 ω 6; C22:4 ω 6; C22:5 ω 3; and C24:1 ω 9) in Group K compared to Group H, while 3 FAs (C18:3 ω 6; C20:1 ω 9; and C22:1 ω 9) showed a decrease in the same group (Table 3). Four FAs of TG (C16:3 ω 4; C18:1 ω 7; C18:2 ω 6; and C24:1 ω 9) increased in Group K whereas 3 FAs (C10:0; C12:0; and C20:3 ω 3) decreased in the same group (Table 4). The differences in CE profiles are shown in Table 5. Among them, 4 FAs (C16:4 ω 1; C20:3 ω 3; C22:5 ω 3; C22:6 ω 3) increased in Group K, while FA C16:1 ω 7 decreased in the same group. 

According to the Boruta test (Figure 1), 9 FAs were used to perform the ROC analysis among, and 3 FAs belonged to PL (C16:2 ω 4; C22:6 ω 3; and C24:1 ω 9), 1 belonged to FFA (C24:1 ω 9), 3 belonged to TG (C12:0; C16:2 ω 4; and C20:3 ω 3), and 2 belonged to CE (C16:1 ω 4; and C22:6 ω 3). The ROC curves identified the following (Figure 2):Only 1 FA was not significant (C16:2 ω 4 TG; *p*-value = 0.063);One FA was a good marker: the C16:2 ω 4 PL (AUC: 0.80, CI: 0.68 to 0.89, cut-off > 0.526, Se: 96%, Sp: 52.6%; *p*-value < 0.001);Six FAs were moderate markers: C24:1 ω 9 FFA (AUC: 0.76, CI: 0.64 to 0.86, cut-off > 0.021, Se: 76%, Sp:79%; *p*-value < 0.001), C20:3 ω 3 TG (AUC: 0.76, CI: 0.63 to 0.86, cut-off ≤ 0.102, Se: 80%, Sp: 68.4%; *p*-value < 0.001), C24:1 ω 9 PL (AUC: 0.74, CI: 0.61 to 0.84, cut-off ≥ 0.374, Se: 72%, Sp: 68.4%; *p*-value < 0.001), C12:0 TG (AUC: 0.74, CI: 0.61 to 0.84, cut-off ≤ 0.522, Se: 56%, Sp: 86.8%; *p*-value < 0.001), C22:6 ω 3 CE (AUC: 0.70, CI: 0.58 to 0.81, cut-off > 0.159, Se: 60%, Sp: 84.2%; *p*-value = 0.007), and C22:6 ω 3 PL (AUC: 0.70, CI: 0.57 to 0.81, cut-off > 0.106, Se: 64%, Sp: 81.6%; *p*-value = 0.008);One FA was a poor marker: C16:4 ω 1 CE (AUC: 0.68, CI: 0.55 to 0.79, cut-off > 0.095, Se: 64%, Sp: 7 9%; *p*-value = 0.023).

## 4. Discussion

The onset of lactation induces greater metabolic demands during the transition period with consequent lipomobilization. The mobilization of lipids from adipose tissue involves the release of FA into the bloodstream. However, this condition not only affects the relative concentrations of total plasma lipids but also shifts the FA composition of the corresponding fractions (PL, FFA, TG, and CE). This change is partially due to the differences in FA distributions among lipid classes. These changes in blood lipids influence energy redistribution and cell metabolism and function [9,19].

The aim of the current study was to assess the lipid fractions in early lactation to investigate the metabolic changes associated with different levels of energy deficit.

The biochemical analysis identified a difference in AST concentrations. This enzyme tends to increase in the presence of liver injuries or muscle damage. Values of about 122 U/L have been associated with hepatic lipidosis in ketotic buffaloes [20,21]. In this study, both groups had slightly higher levels that might suggest the presence of hepatic damage.

Each lipid class has different metabolic functions. The PLs are the main components of biological membranes [9]. Furthermore, the FA composition of plasma PL is well correlated with the cell membrane FA composition of leukocytes in cows [22]. The main FAs of this lipid class are linoleic (C18:2 ω 6), stearic (C18:0), palmitic (C16:0), and oleic (C18:1 ω 9) acids in dairy cows [23]. This was also confirmed in the present study of buffalo plasma lipid classes, in which these four FAs accounted around 83% of the PLFA in both groups. Although there was no difference between groups in PL or PLFA concentrations, there was a difference in their profile. The identified changes involve increased concentrations of caprylic (C8:0), hexadecadienoic (C16:2 ω 4), oleic, vaccenic (C18:1 ω 7), gadoleic (C20:1 ω 11), gondoic (C20:1 ω 9), docosahexaenoic (DHA; C22:6 ω 3), and nervonic (C24:1 ω 9) acids in Group K. Even if only one saturated FA is increased, it can negatively affect membrane fluidity and, consequently, cellular function [24]. In addition, changes in the ω-3 profile of the membrane, such as DHA, may influence immune responses by leukocytes. Specifically, a decreased concentration of ω-3 associated with an increased concentration of palmitic acid may enhance leukocyte activation [19]. Nevertheless, palmitic acid was not found to be modified, and DHA increased, suggesting the absence of leukocyte activation. Anyway, this study used a cross-sectional experimental design; therefore, knowing the exact previous state of the animals is not possible. Moreover, five of eight increased FAs among PLFA were mono-unsaturated fatty acids (MUFAs). The MUFA content in PL is highly dependent on the gene expression of lipid-metabolizing enzymes such as stearoyl-desaturase 1 (SCD1) [25]. The expression of this gene with others was found to be increased in the liver of ketotic cows compared to subclinical or control animals and relevant for hepatic lipidosis [26]. Considering the increment of MUFA in the PL class and that Group K was only at risk of ketosis, these results suggest that more investigations of metabolic conditions in buffalo is needed because the manifestation of ketosis in this species may differ from cows.

The FFA is mainly transported in the bloodstream linked to albumin, and only a small amount is unbound. These FA are usually used for readily complete oxidation by a variety of tissues [9]. Ten FFAs were different between groups: γ-linolenic (GLA; C18:3 ω 6), gondoic (C20:1 ω 9), and erucic (C22:1 ω 9) acids decreased in Group K, whereas hexadecatetraenoic (C16:4 ω 1), eicosatrienoic (ETE; C20:3 ω 3), eicosapentanoic (EPA; C20:5 ω 3), docosadienoic (C22:2 ω 6), adrenic acid (ADA; C22:4 ω 6), docosapentaenoic (DPA; C22:5 ω 3), and nervonic (C24:1 ω 9) acids increased in the same group. Three increased FAs (ETE, EPA, and DPA) were metabolically related to each other and a component of ω-3 FA, which may be mobilized during inflammation due to their derived anti-inflammatory or pro-resolving mediators [15,27]. Also related to the inflammatory response, there were ω-6 FAs, such as docosadienoic acid, ADA, and GLA. ω-6 FAs can be used for the production of both pro- and anti-inflammatory mediators [19]. In particular, docosadienoic acid showed a positive relationship with inflammatory and oxidative stress states in horses [28]. In contrast, GLA and ADA can be used to synthesize arachidonic acid using elongase and desaturase or β-oxidation, respectively [29,30]. Increased docosadienoic acid levels could suggest an inflammatory state and reduced GLA levels that are potentially linked to the synthesis or maintenance of arachidonic acids with concentrations that do not show significant differences between the groups. However, ADA levels showed an increase in Group K. The cross-sectional design did not allow for assessing the FA profile over time but allowed for assessing the development or resolution of the inflammatory state. Furthermore, changes in MUFA concentrations, such as nervonic, gondoic, and erucic acids, may also be related to changes in SCD1 expression and the PL class. These results suggest an influence of the FFA profile that is available for immediate oxidation and potentially related to inflammatory states, but further studies over time are needed to further investigate these aspects in buffaloes. 

TG represents the main storage site of long-chain FA in adipose or mammary tissues or the blood, such as very low-density lipoproteins (VLDLs), which represent their main carrier in the bloodstream [8,10]. The TG concentrations in the VLDL and liver are related to each other in dairy cows [31]. In fact, deposits in adipose tissue are mobilized with the onset of lactation to support increased energy demands. The influx of FA to the liver induces the increased synthesis of TG, which is normally exported from the liver as part of VLDL. However, if TG synthesis exceeds hepatic export capacity, it will accumulate in the vesicles in hepatocytes, leading to a fatty liver [10,32,33]. This lipid fraction usually shows an increase in palmitic, oleic, and linoleic acids during lipomobilization, while PUFA shows a depletion [34,35]. In our study, changes in palmitic or oleic acids were not evidenced. Instead, an increased concentration of linoleic acid (C18:2 ω 6) was evidenced in Group K. Regarding the PUFA level, a difference was present for hexadecatrienoic acid (C16:3 ω 4) and ETE (C20:3 ω 3), for which their concentrations increased and reduced in Group K, respectively. The decreased ETE level may be due to its mobilization as an FFA and its use for modulating inflammatory responses as previously mentioned. The study by Peter et al. [31] on humans reported that the TG fraction of VLDL is related to SCD1 expression in the liver. In our study, increased levels of vaccenic (C18:1 ω 7) and nervonic (C24:1 ω 9) acids were present in Group K. Anyway, further studies that focused on ruminants, especially buffaloes, should be performed to better assess this relationship. 

The activity of lecithin-cholesterol acyltransferase (LCAT) enzymes and CE concentrations commonly exhibit low levels during the onset of lactation with the presence of a negative energy balance. However, these levels tend to increase progressively toward mid-lactation due to the metabolic adaptation of animals [36,37]. In our study, no difference in CE levels was identified between groups. Anyway, a different profile was present and particularly related to the increased level of ω-3 FA. CEs are generally found in membranes and lipoproteins. In the latter case, their function together with TGs is lipid transport between tissues [10]. In this lipid class, the concentration of ω-3 FA is expected to decrease in the first week of lactation due to the inflammatory response [19]. Considering that the animals in our study are in early lactation, the increased levels of ω-3 FA in Group K could indicate a progressive restoration of body lipid reserves, or their use has not yet occurred considering that they were still readily available as FFA. In addition, the increased levels of ω-3 FA in circulating FFAs may have also affected the profile of CE synthesized in the liver. 

Among the significant FA in each lipid class, some FAs were selected using the Boruta test for their predictive function, and they were confirmed in eight out of nine FAs via ROC analysis. Among these FA, there were one saturated FA, two MUFAs, and five PUFAs from which three were ω-3 FAs. These FAs probably showed a greater biological significance in discriminations according to BHB levels and, therefore, should be evaluated more with respect to their specific physio-pathological functions. In particular, hexadecadienoic acid (C16:2 ω 4 PL) seems to show a clear promising role in MBs. Its biological function has been in-depth studied in human medicine, which awarded it with anti-inflammatory properties [38]. Indeed, it has been demonstrated that hexadecanoic acid can act as an anti-inflammatory agent by competing with phospholipase A2 and consequently inhibiting its activity [38]. This enzyme can release FA as arachidonic acid from cell membrane PLs via the hydrolysis of the ester bond at the sn-2 position. This enzymatic reaction is an important step in the formation of inflammatory mediators [38].

Our results showed higher concentrations of hexadecadienoic acid in buffaloes at risk of ketosis (Group K), enabling the authors to hypothesize an active role in the anti-inflammatory response and the potential development of distress and diseases related to a state of energy deficit. Looking at this interesting finding, its role in buffaloes should receive further attention from studies aiming to assess its reliability as a predictive marker of impaired metabolic statuses and related diseases.

## 5. Conclusions

Changes in the plasma FA profiles of major lipid classes may help further investigate metabolic alterations and adaptations in buffaloes during early lactation. Increased BHB levels above 0.70 mmol/L are associated with altered lipid class profiles. These highlighted changes among the FA profiles of lipid classes suggest the influence of inflammatory responses, liver metabolism, and body lipid reserve statuses. However, the cross-sectional experimental design limited the possibility of exactly assessing inflammation and body lipid reserve statuses. In addition, a suspected alteration of liver gene expression was hypothesized, and downregulation was evidenced in ketotic cows. Accordingly, the possible similarities in buffaloes at risk of hyperketonemia and ketotic cows suggest further investigations of this metabolic disease to evaluate a specific threshold of BHB and the manifestation of ketosis in this species.

## Figures and Tables

**Figure 1 animals-13-02333-f001:**
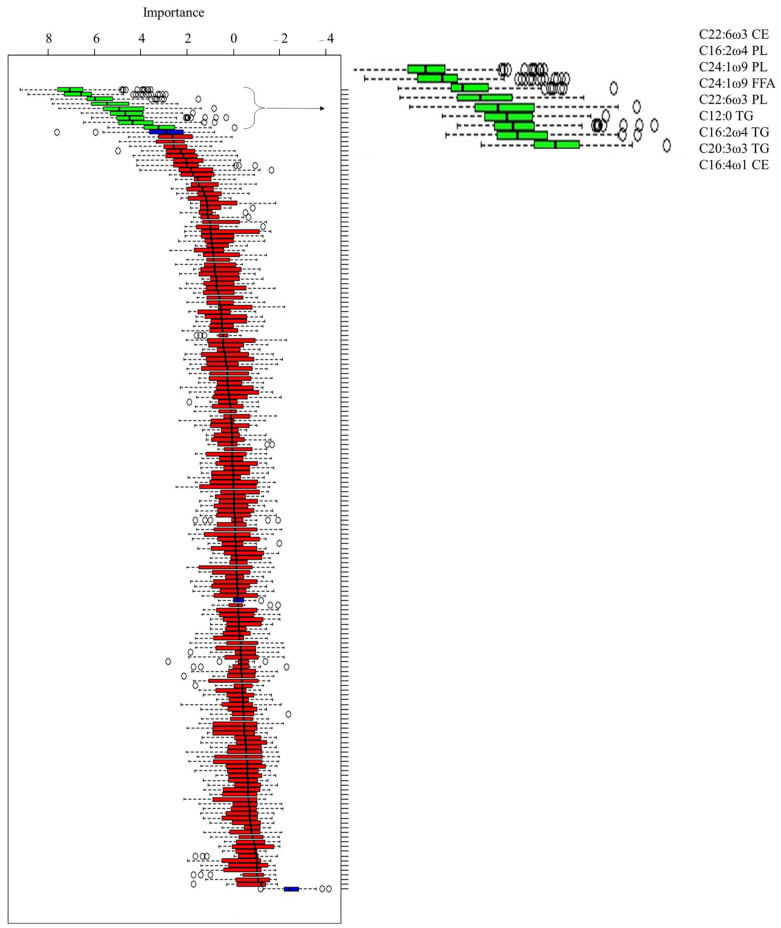
Box plot of the Boruta decisional algorithm. The green box plot represents the predictive fatty acids selected by the Boruta analysis. The blue box plot represents fatty acids with a doubtful predictive function, while the red box plot represents fatty acids with a null predictive function.

**Figure 2 animals-13-02333-f002:**
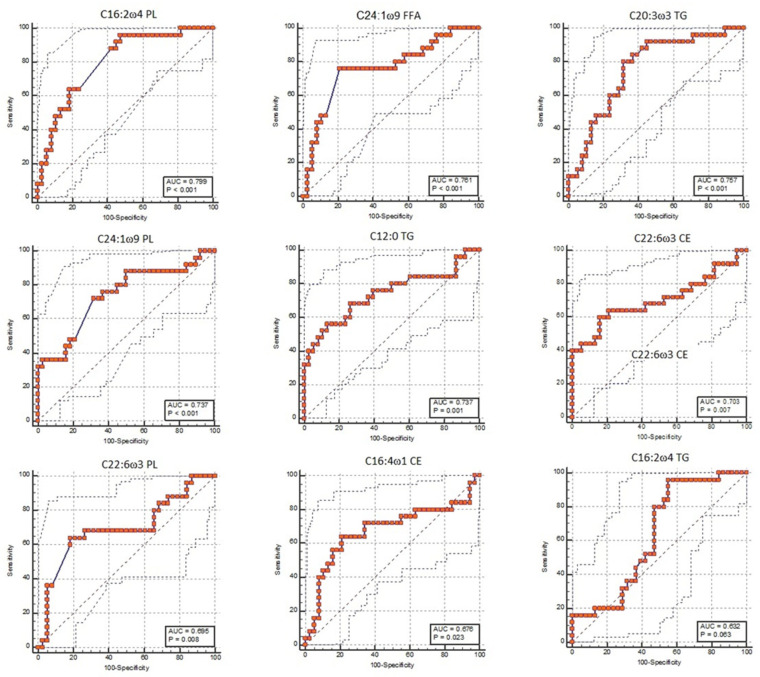
Receiver operating characteristic (ROC) curve of predictive fatty acids listed with a decreasing predictive function (area under the curve (AUC)) and obtained using the Boruta decisional algorithm.

**Table 1 animals-13-02333-t001:** Animal data and biochemical parameters of MBs divided into Group H (healthy group; BHB < 0.70 mmol/L) and Group K (at risk of hyperketonemia; BHB ≥ 0.70 mmol/L).

Parameters	Group H (*n* = 38)	Group K (*n* = 25)	SEM	*p*-Values
BCS ^1^	4.58	5.08	0.18	0.058
Parity	3.53	3.89	0.37	0.506
DIM ^2^	30.4	33.5	2.19	0.330
Milk production (Kg/d)	14.3	14.9	0.58	0.461
BHB ^3^ (mmol/L)	0.47	0.74	0.02	<0.001
NEFA ^4^ (mEq/L)	0.24	0.25	0.02	0.883
CHO ^5^ (mg/dL)	75.0	86.2	5.48	0.161
TGR ^6^ (mg/dL)	9.35	10.09	0.49	0.302
GLU ^7^ (mg/dL)	62.4	64.3	1.28	0.313
GGT ^8^ (U/L)	19.6	21.8	1.06	0.170
AST ^9^ (U/L)	140.0	164.0	7.12	0.017
ALT ^10^ (U/L)	48.0	49.4	2.07	0.638

^1^ Body condition score; ^2^ days in milk; ^3^ β-Hydroxybutyrate; ^4^ non-esterified fatty acid; ^5^ cholesterol; ^6^ triglycerides; ^7^ glucose; ^8^ γ-Glutamyl-transferase; ^9^ aspartate transaminase; ^10^ alanine transaminase.

**Table 2 animals-13-02333-t002:** Least square means and standard error of the mean (SEM) of plasma phospholipid (PL) profiles associated with Group H (healthy group; BHB < 0.70 mmol/L) and Group K (at risk of hyperketonemia; BHB ≥ 0.70 mmol/L).

PL	Nomenclature	Group H (*n* = 38)	Group K (*n* = 25)	SEM	*p*-Values
C6:0	Caproic acid	0.05	0.06	0.004	0.510
C8:0	Caprylic acid	0.03	0.05	0.005	0.017
C10:0	Capric acid	0.05	0.05	0.006	0.650
C12:0	Lauric acid	0.41	0.46	0.068	0.750
C14:0	Myristic acid	0.43	0.47	0.045	0.760
C14:1 ω 5	Myristelaidic acid	0.14	0.17	0.016	0.560
C16:0	Palmitic acid	15.8	17.1	0.868	0.840
C16:1 ω 9	Hypogeic acid	0.15	0.16	0.010	0.810
C16:1 ω 7	Palmitoleic acid	0.32	0.38	0.022	0.290
C16:2 ω 4	Hexadecadienoic acid	0.43	1.24	0.121	<0.001
C16:3 ω 4	Hexadecatrienoic acid	0.03	0.04	0.003	0.140
C16:4 ω 1	Hexadecatetraenoic acid	0.04	0.04	0.006	0.400
C17:0	Margaric acid	0.84	0.85	0.060	0.850
C17:1 ω 7	Heptadecenoic acid	0.14	0.15	0.011	0.830
C18:0	Stearic acid	18.2	20.6	1.125	0.380
C18:1 ω 9	Oleic acid	9.15	10.9	0.508	0.059
C18:1 ω 7	Vaccenic acid	1.13	1.47	0.092	0.020
C18:2 ω 6	Linoleic acid	20.0	22.3	1.515	0.750
C18:3 ω 6	γ-Linolenic acid (GLA)	0.12	0.14	0.011	0.360
C18:3 ω 3	α-Linolenic acid (ALA)	0.97	1.01	0.081	0.910
C18:4 ω 3	Stearidonic acid (SDA)	0.01	0.01	0.002	0.710
C20:0	Arachidic acid	0.12	0.13	0.006	0.310
C20:1 ω 11	Gadoleic acid	0.06	0.08	0.006	0.083
C20:1 ω 9	Gondoic acid	0.04	0.05	0.004	0.021
C20:2 ω 6	Eicosadienoic acid	0.11	0.12	0.009	0.420
C20:3 ω 9	Mead acid	0.26	0.27	0.030	0.850
C20:3 ω 6	Dihomo-γ-Linolenic acid (DGLA)	1.44	1.64	0.108	0.440
C20:4 ω 6	Arachidonic acid	2.41	2.62	0.162	0.620
C20:3 ω 3	Eicosatrienoic acid (ETE)	0.03	0.03	0.003	0.480
C20:4 ω 3	Eicosatetranoic acid (ETA)	0.09	0.11	0.008	0.190
C20:5 ω 3	Eicosapentanoic acid (EPA)	0.54	0.65	0.044	0.250
C22:0	Behenic acid	0.47	0.54	0.024	0.170
C22:1 ω 9	Erucic acid	0.01	0.01	0.001	0.510
C22:1 ω 7	15-Docosenoic acid	0.01	0.01	0.001	0.470
C22:2 ω 6	Docosadienoic acid	0.05	0.05	0.004	0.530
C22:4 ω 6	Adrenic acid (ADA)	0.29	0.33	0.027	0.660
C22:5 ω 3	Docosapentaenoic acid (DPA)	0.61	0.65	0.041	0.690
C22:6 ω 3	Docosahexaenoic acid (DHA)	0.09	0.23	0.034	0.003
C24:0	Lignoceric acid	0.50	0.55	0.038	0.560
C24:1 ω 9	Nervonic acid	0.26	0.44	0.035	<0.001
PLFA mg/dL	/	76.0	85.9	4.47	0.420
PL mg/dL	/	105.0	119.0	6.18	0.410

**Table 3 animals-13-02333-t003:** Least square means and standard error of the mean (SEM) of plasma free fatty acid (FFA) profiles associated with Group H (healthy group; BHB < 0.70 mmol/L) and Group K (at risk of hyperketonemia; BHB ≥ 0.70 mmol/L).

FFA	Nomenclature	Group H (*n* = 38)	Group K (*n* = 25)	SEM	*p*-Values
C6:0	Caproic acid	0.03	0.03	0.002	0.450
C8:0	Caprylic acid	0.02	0.02	0.003	0.670
C10:0	Capric acid	0.03	0.03	0.003	0.750
C12:0	Lauric acid	0.37	0.33	0.026	0.540
C14:0	Myristic acid	0.19	0.20	0.020	0.680
C14:1 ω 5	Myristelaidic acid	0.02	0.01	0.001	0.240
C15:0	Pentadecanoic Acid	0.06	0.07	0.005	0.690
C16:0	Palmitic acid	1.37	1.42	0.097	0.920
C16:1 ω 9	Hypogeic acid	0.02	0.02	0.002	0.300
C16:1 ω 7	Palmitoleic acid	0.04	0.04	0.006	0.940
C16:2 ω 4	Hexadecadienoic acid	0.01	0.01	0.001	0.300
C16:3 ω 4	Hexadecatrienoic acid	0.004	0.003	0.001	0.540
C16:4 ω 1	Hexadecatetraenoic acid	0.010	0.012	0.001	0.080
C17:1	Heptadecenoic acid	0.01	0.01	0.002	0.730
C18:0	Stearic acid	1.68	0.78	0.113	0.770
C18:1 ω 9	Oleic acid	0.76	0.82	0.117	0.910
C18:1 ω 7	Vaccenic acid	0.06	0.07	0.007	0.480
C18:2 ω 6	Linoleic acid	0.12	0.12	0.012	0.670
C18:3 ω 6	γ-Linolenic acid (GLA)	0.042	0.039	0.001	0.030
C18:3 ω 3	α-Linolenic acid (ALA)	0.01	0.01	0.001	0.280
C18:4 ω 3	Stearidonic acid (SDA)	0.004	0.004	0.001	0.840
C20:0	Arachidic acid	0.02	0.02	0.001	0.940
C20:1 ω 11	Gadoleic acid	0.004	0.005	0.001	0.190
C20:1 ω 9	Gondoic acid	0.006	0.004	0.001	0.016
C20:2 ω 6	Eicosadienoic acid	0.005	0.004	0.001	0.320
C20:3 ω 9	Mead acid	0.003	0.003	0.001	0.630
C20:3 ω 6	Dihomo-γ-Linolenic acid (DGLA)	0.01	0.01	0.001	0.520
C20:4 ω 6	Arachidonic acid	0.01	0.01	0.001	0.670
C20:3 ω 3	Eicosatrienoic acid (ETE)	0.007	0.008	0.0001	0.095
C20:4 ω 3	Eicosatetranoic acid (ETA)	0.004	0.003	0.001	0.220
C20:5 ω 3	Eicosapentanoic acid (EPA)	0.02	0.04	0.006	0.039
C22:0	Behenic acid	0.01	0.01	0.001	0.150
C22:1 ω 9	Erucic acid	0.004	0.002	0.001	0.088
C22:1 ω 7	15-Docosenoic acid	0.003	0.002	0.001	0.490
C22:2 ω 6	Docosadienoic acid	0.016	0.019	0.001	0.044
C22:4 ω 6	Adrenic acid (ADA)	0.02	0.03	0.003	0.018
C22:5 ω 3	Docosapentaenoic acid (DPA)	0.01	0.27	0.070	0.018
C22:6 ω 3	Docosahexaenoic acid (DHA)	0.02	0.06	0.053	0.590
C24:0	Lignoceric acid	0.02	0.03	0.006	0.120
C24:1 ω 9	Nervonic acid	0.02	0.04	0.005	0.001
FFA mg/dL		4.99	5.36	0.369	0.740

**Table 4 animals-13-02333-t004:** Least square means and standard error of the mean (SEM) of plasma triglyceride (TG) profiles associated with Group H (healthy group; BHB < 0.70 mmol/L) and Group K (at risk of hyperketonemia; BHB ≥ 0.70 mmol/L).

TG	Nomenclature	Group H (*n* = 38)	Group K (*n* = 25)	SEM	*p*-Values
C6:0	Caproic acid	0.03	0.03	0.002	0.890
C8:0	Caprylic acid	0.04	0.05	0.006	0.320
C10:0	Capric acid	0.07	0.05	0.006	0.036
C12:0	Lauric acid	0.76	0.54	0.063	0.025
C14:0	Myristic acid	0.26	0.19	0.040	0.240
C14:1 ω 5	Myristelaidic acid	0.02	0.03	0.001	0.380
C15:0	Pentadecanoic Acid	0.17	0.18	0.016	0.840
C16:0	Palmitic acid	2.28	2.20	0.117	0.360
C16:1 ω 9	Hypogeic acid	0.03	0.04	0.005	0.260
C16:1 ω 7	Palmitoleic acid	0.04	0.04	0.004	1.000
C16:2 ω 4	Hexadecadienoic acid	0.04	0.04	0.004	0.120
C16:3 ω 4	Hexadecatrienoic acid	0.004	0.01	0.001	0.014
C16:4 ω 1	Hexadecatetraenoic acid	0.01	0.02	0.001	0.450
C17:1 ω 7	Heptadecenoic acid	0.01	0.01	0.001	1.000
C18:0	Stearic acid	3.53	3.98	0.244	0.280
C18:1 ω 9	Oleic acid	0.46	0.53	0.053	0.470
C18:1 ω 7	Vaccenic acid	0.18	0.26	0.024	0.025
C18:2 ω 6	Linoleic acid	0.20	0.44	0.101	0.095
C18:3 ω 6	γ-Linolenic acid (GLA)	0.05	0.05	0.001	0.830
C18:3 ω 3	α-Linolenic acid (ALA)	0.01	0.01	0.001	0.450
C18:4 ω 3	Stearidonic acid (SDA)	0.01	0.01	0.001	0.120
C20:0	Arachidic acid	0.06	0.06	0.004	0.630
C20:1 ω 11	Gadoleic acid	0.01	0.12	0.002	0.150
C20:1 ω 9	Gondoic acid	0.01	0.01	0.001	0.101
C20:2 ω 6	Eicosadienoic acid	0.004	0.001	0.001	0.340
C20:3 ω 9	Mead acid	0.004	0.005	0.001	0.260
C20:3 ω 6	Dihomo-γ-Linolenic acid (DGLA)	0.02	0.02	0.001	0.910
C20:4 ω 6	Arachidonic acid	0.01	0.01	0.002	0.220
C20:3 ω 3	Eicosatrienoic acid (ETE)	0.11	0.09	0.004	0.001
C20:4 ω 3	Eicosatetranoic acid (ETA)	0.01	0.01	0.001	0.400
C20:5 ω 3	Eicosapentanoic acid (EPA)	0.02	0.03	0.005	0.580
C22:0	Behenic acid	0.03	0.04	0.002	0.130
C22:1 ω 9	Erucic acid	0.003	0.004	0.001	0.530
C22:1 ω 7	15-Docosenoic acid	0.004	0.004	0.001	0.410
C22:2 ω 6	Docosadienoic acid	0.01	0.01	0.001	0.560
C22:4 ω 6	Adrenic acid (ADA)	0.01	0.01	0.001	0.160
C22:5 ω 3	Docosapentaenoic acid (DPA)	0.01	0.01	0.001	0.930
C22:6 ω 3	Docosahexaenoic acid (DHA)	0.01	0.01	0.002	0.520
C24:0	Lignoceric acid	0.03	0.04	0.003	0.130
C24:1 ω 9	Nervonic acid	0.04	0.10	0.012	0.002
TGFA mg/dL	/	8.59	9.12	0.51	0.600
TG mg/dL	/	9.03	9.58	0.54	0.600

**Table 5 animals-13-02333-t005:** Least square means and standard error of the mean (SEM) of plasma cholesterol ester (CE) profiles associated with Group H (healthy group; BHB < 0.70 mmol/L) and Group K (at risk of hyperketonemia; BHB ≥ 0.70 mmol/L).

CE	Nomenclature	Group H (*n* = 38)	Group K (*n* = 25)	SEM	*p*-Values
C6:0	Caproic acid	0.06	0.07	0.004	0.160
C8:0	Caprylic acid	0.08	0.06	0.021	0.630
C10:0	Capric acid	0.14	0.12	0.012	0.400
C12:0	Lauric acid	0.73	0.65	0.075	0.610
C14:0	Myristic acid	0.61	0.57	0.077	0.480
C14:1 ω 5	Myristelaidic acid	0.11	0.12	0.006	0.990
C15:0	Pentadecanoic Acid	0.49	0.50	0.050	0.700
C16:0	Palmitic acid	8.52	8.81	0.594	0.760
C16:1 ω 9	Hypogeic acid	0.69	0.90	0.142	0.560
C16:1 ω 7	Palmitoleic acid	1.41	1.11	0.109	0.012
C16:2 ω 4	Hexadecadienoic acid	0.17	0.18	0.020	0.810
C16:3 ω 4	Hexadecatrienoic acid	0.14	0.16	0.019	0.680
C16:4 ω 1	Hexadecatetraenoic acid	0.09	0.12	0.010	0.024
C17:1 ω 7	Heptadecenoic acid	0.54	0.61	0.040	0.410
C18:0	Stearic acid	1.88	1.93	0.907	0.890
C18:1 ω 9	Oleic acid	18.1	19.8	0.161	0.470
C18:1 ω 7	Vaccenic acid	0.66	0.61	0.052	0.280
C18:2 ω 6	Linoleic acid	42.7	45.4	3.405	0.860
C18:3 ω 6	γ-Linolenic acid (GLA)	1.85	1.86	0.200	0.600
C18:3 ω 3	α-Linolenic acid (ALA)	1.72	1.85	0.118	0.980
C18:4 ω 3	Stearidonic acid (SDA)	0.04	0.04	0.004	0.340
C20:0	Arachidic acid	0.05	0.05	0.011	0.780
C20:1 ω 11	Gadoleic acid	0.10	0.10	0.012	0.780
C20:1 ω 9	Gondoic acid	0.04	0.05	0.005	0.420
C20:2 ω 6	Eicosadienoic acid	0.20	0.20	0.005	0.570
C20:3 ω 9	Mead acid	0.07	0.07	0.007	0.970
C20:3 ω 6	Dihomo-γ-Linolenic acid (DGLA)	0.27	0.30	0.023	0.540
C20:4 ω 6	Arachidonic acid	0.94	0.98	0.065	0.960
C20:3 ω 3	Eicosatrienoic acid (ETE)	0.11	0.13	0.005	0.029
C20:4 ω 3	Eicosatetranoic acid (ETA)	0.10	0.10	0.005	0.700
C20:5 ω 3	Eicosapentanoic acid (EPA)	0.49	0.51	0.026	0.940
C22:0	Behenic acid	0.210	0.22	0.015	0.700
C22:1 ω 9	Erucic acid	0.05	0.07	0.009	0.330
C22:1 ω 7	15-Docosenoic acid	0.04	0.04	0.009	0.460
C22:2 ω 6	Docosadienoic acid	0.04	0.03	0.015	0.160
C22:4 ω 6	Adrenic acid (ADA)	0.06	0.06	0.013	0.970
C22:5 ω 3	Docosapentaenoic acid (DPA)	0.14	0.68	0.134	0.010
C22:6 ω 3	Docosahexaenoic acid (DHA)	0.10	0.23	0.021	<0.001
C24:0	Lignoceric acid	0.70	0.60	0.082	0.150
C24:1 ω 9	Nervonic acid	0.16	0.16	0.016	0.640
CEFA mg/dL	/	84.4	89.7	5.25	0.950
CE mg/dL	/	197.0	210.0	12.25	0.940

## Data Availability

The data will be available upon request by sending an email to the corresponding author.

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
