# Peer review of "Metabolic Changes Associated with Different Levels of Energy Deficits in Mediterranean Buffaloes during the Early Lactation Stage: Type and Role of the Main Lipid Fractions Involved"

_animals, 2023, doi:10.3390/ani13142333_

Round 1

Reviewer 1 Report

The presented study is very complex and interesting. It contains quite a large number of results that can help optimize the management of metabolism in buffaloes.

I would add to the objective of the thesis that the analyzes were performed from milk.

Please explain what are the short-circuits in the Boruta test results - AUC, CI, Se and Sp.

It would be advisable to describe at least some climatic conditions in the analysis of the farm.

Another recommendation is to describe how large a herd the animals were selected from and the housing system should be at least basically characterized. The order of calving should also be added.

Either the amount of blood taken does not match, or the blood was taken more than once - drops for BHB, 3 ml for EDTA and 9 ml for a tube with an activator.

P-values < 0.0001 are usually only reported as < 0.001 in the results.

Please what does index 5 on line 299 and 7 on line 342 mean.

Bad form of citation (Peter et al. 2009) on line 329.

Miss "." at the end of the conclusion.

I would appreciate a precisely described discussion with all the metabolic relationships. However, the conclusion is quite short and I would recommend expanding it a bit. The work is of a high standard and after minor adjustments I recommend it for publication.

English is at a high level with only minor shortcomings.

Author Response

Reviewer 1

The presented study is very complex and interesting. It contains quite a large number of results that can help optimize the management of metabolism in buffaloes.

  1. I would add to the objective of the thesis that the analyzes were performed from milk.

AU: Thank you for your suggestion. Samples used for TLC-GC analysis are plasma derived from blood.

We changed lines 67-71 “The plasma lipid fractions of buffaloes related to ketone bodies has not been investigated, to the best of the authors’ knowledge. The aim of the current study was to use gas chromatography associated with thin layer chromatography (TLC-GC) to assess the lipid fractions of Mediterranean buffaloes (MBs) in early lactation to investigate the metabolic changes associated with different levels of energy deficit.”

In

" To the best of the authors’ knowledge, the blood plasmatic lipid fractions related to ketone bodies have not been investigated in buffaloes. The aim of this study was to investigate metabolic alterations related to changes in plasma lipid classes according to different level of energy deficit in early lactating Mediterranean buffaloes (MBs) using gas chromatography associated with thin layer chromatography (TLC-GC)."

  1. Please explain what are the short-circuits in the Boruta test results - AUC, CI, Se and Sp.

AU: Thank you for your suggestion. If we understood correctly, we changed lines 189-193 “The area under the curve (AUC) expresses the diagnostic power of the test as follow: AUC of 0.9 to 1.0 = excellent, 0.8 to 0.9 = good, 0.7 to 0.8 = moderate, 0.6 to 0.7 = poor, and 0.5 to 0.6 = fail [17].”

In

“The diagnostic power was assessed through the area under the curve (AUC) that identify a FA as an excellent marker when the AUC is 0.9 to 1.0; good if AUC is 0.8 to 0.9; moderate if AUC is 0.7 to 0.8; poor if AUC is 0.6 to 0.7, and fail if AUC is 0.5 to 0.6 [18]. The AUC is associated with the 95% confidence interval (CI), the sensitivity (Se) and specificity (Sp) of the test.”

Line 222, we deleted “Confidence Interval (CI)”

Lines 178-182, we added “Boruta algorithm is a random forest classification algorithm that provides a numerical estimate of the feature importance. During this analysis, multiple unbiased weak classifiers (decision trees) are used to perform a classification independently between decision trees parts. The importance measure of classifiers is obtained as the loss of accuracy of classification. Then the average and standard deviation of the accuracy loss are computed”

  1. It would be advisable to describe at least some climatic conditions in the analysis of the farm.

AU: thank you for your suggestion, the information regarding the timing of sampling was added (months in which was performed: line 80 “January and April 2019”). The latter information combined with the place of the study (southern Italy) should give enough information to climatically characterize the area of the study (line 75).

  1. Another recommendation is to describe how large a herd the animals were selected from and the housing system should be at least basically characterized. The order of calving should also be added.

AU: thank you for your recommendation, the number of buffalo heads was already reported within the manuscript between the selection criteria (Please, see line 84) while additional information regarding the housing systems has been added in lines, 88-90: “The selected barn was characterized with solid grooved concrete floors in the walking and feeding alleys. The lying area was whether made by elevated cubicles covered with rubber mattresses (for milking MBs) or a roofed deep straw yard area (for dry MB).”

Unfortunately, the reviewer's request regarding calving is not fully clear to the authors. If the Reviewer is referring to the type of calving system, the information has been added to the manuscript: lines 75-76, we added “and characterized by an artificially induced seasonal calving herd (late winter-springtime)”.

If you are referring to the order of calving of the enrolled animals, as reported within the manuscript the Authors enrolled all the buffalo within 50 DIM (Additional details in Table 1) during a defined time-recruitment (Please see line 80, “January and April 2019”).

  1. Either the amount of blood taken does not match, or the blood was taken more than once - drops for BHB, 3 ml for EDTA and 9 ml for a tube with an activator.

AU:  Thank you for your questions. Blood was sampled with a 10 mL syringe as long as it was possible to pull the plunger (larger sized syringes would have been slightly more difficult for the time of sampling). The tubes were therefore not filled to the top, but the samples were still sufficient to obtain a quantity of blood to be able to perform all the following analyses.

To avoid misunderstanding in the reader, we have removed references to maximum tube capacity (lines 109 and 110).

  1. P-values < 0.0001 are usually only reported as < 0.001 in the results.

AU: Thank you for your suggestion. We changed the manuscript and tables as suggested.

  1. Please what does index 5 on line 299 and 7 on line 342 mean.

AU: Thank you for your questions. We apologize, this is a mistake in the process of references editing according to the guidelines. We have removed these indexes (lines 305 and 348).

  1. Bad form of citation (Peter et al. 2009) on line 329.

AU: Thank you for your questions. If we have correctly understood what the bad form indicated is, we have changed line 335, “The study of Peter et al. (2009) [30]”

In

“The study of Peter et al. [30]”

  1. Miss "."at the end of the conclusion.

AU: Thank you for your suggestion. We added to point.

  1. I would appreciate a precisely described discussion with all the metabolic relationships.However, the conclusion is quite short and I would recommend expanding it a bit.The work is of a high standard and after minor adjustments I recommend it for publication.

AU: Thank you for your questions. Lines 374-384, we changed “Changes in plasma FA profiles of major lipid classes may help to further investigate metabolic alterations and adaptations in buffaloes during early lactation. The highlighted changes among FA profiles of lipid classes suggest possible relationships with lipid metabolism and inflammation. In addition, possible similarities in buffaloes at risk of hyperketonemia with ketotic cows suggest further investigations of ketosis in buffaloes are needed because manifestation seems to differ between the two species”

In

“Changes in plasma FA profiles of major lipid classes may help to further investigate metabolic alterations and adaptations in buffaloes during early lactation. Increased BHB levels above 0.70 mmol/L are associated with altered lipid class profiles. These highlighted changes among FA profiles of lipid classes suggest the influence of inflammatory response, liver metabolism, and body lipid reserves status. However, the cross-sectional experimental design limited the possibility of exactly assessing inflammation and body lipid reserves status. In addition, a suspected alteration of liver gene expression was hypothesized, which downregulation was evidences in ketotic cows. Accordingly, the possible similarities in buffaloes at risk of hyperketonemia and ketotic cows suggest further investigations of this metabolic disease to evaluate a specific threshold of BHB and the manifestation of ketosis in this species.”

Reviewer 2 Report

Our thoughts on the article entitled “Metabolic changes associated with different levels of energy deficit in Mediterranean Buffaloes at early lactation stage: type and role of the main lipid fractions involved“.

The article seems to be interesting as it deals mainly with the composition of different lipids in Bubalus bubalis.

But we do make some considerations.

In lines 26-27 it is said that changes in metabolism and inflammation have been identified.  How were metabolic changes identified? Which indicators of inflammation were used?  The ω 3 and ω 6 fatty acid profile does not guarantee an inflammatory state. Acute phase protein analysis should have been performed.

On line 84, the absence of a regular monitoring program for metabolic diseases, did not interfere with the results?

Which were the BCS used in the groups at the beginning of the experiment? How did it vary within the groups from the beginning of the experiment until the end? Lines 96 - 100

In lines 113 – 115, are the plasma/serum amounts for the pyrogallol amount, correct? “For each plasma sample, 250 µL of plasma was immediately transferred to Eppendorf tubes (1mL of serum/tube) containing 5 mg of pyrogallol to reduce FA oxidation “;

Explain in line 114 the statement (1mL of serum/tube); was 250 μL of plasma added to a microtube that already contained 1 mL of serum?

The objectives emphasized in three moments, (lines 67 – 70; lines 24-26; lines 259-260) speak to energy levels and metabolic changes; How were energy levels determined? It is not described in the text. What metabolic changes have been identified? Very broad statement.

In the Abstract the results are not mentioned as proposed in the objectives, the metabolic alterations associated with different levels of energy deficit, only that there was a change in lipid classes and suggests a relationship in lipid metabolism and inflammation.

It is important to note that we did not see in the paper how different energy deficits were identified in the animals; How to recognize that animals are at different levels of energy deficit?

In lines 86 – 92, were the animals of the different groups fed the same amount?

In line 89, how much sodium bicarbonate?

In the conclusions (lines 367 – 373), it is stated that there are possible similarities in buffaloes at risk of hyper ketonemia with ketotic cows and suggests further investigation of ketosis in buffaloes because manifestation seems to differ between the two species; Unclear.

Author Response

Our thoughts on the article entitled “Metabolic changes associated with different levels of energy deficit in Mediterranean Buffaloes at early lactation stage: type and role of the main lipid fractions involved “.

The article seems to be interesting as it deals mainly with the composition of different lipids in Bubalus bubalis, but we do make some considerations.

  1. In lines 26-27 it is said that changes in metabolism and inflammation have been identified. How were metabolic changes identified? Which indicators of inflammation were used?  The ω 3 and ω 6 fatty acid profile does not guarantee an inflammatory state. Acute phase protein analysis should have been performed.

AU: Thank you for your questions. Different energy demands during the onset of lactation in which a NEB state is present induce lipomobilization with changes in the composition of major lipid classes. These changes thus in themselves represent a change in the animal's metabolism, which must adapt to the different energy demands (Contreras et al., 2010; Horst et al., 2021). In this context of metabolic changes, our results suggested an altered inflammatory response. However, the cross-sectional experimental design does not allow us to understand exactly at what stage of inflammation the animals are, as also explained in lines 314-320. Certainly, the acute phase proteins may help in improving the identification of the animals, but unfortunately it was not part of the initial experimental design. We would, therefore, like to thank the reviewer for providing us with ideas to improve our studies. Anyway, the acute phase protein response depends on the release of signal molecules including products of ω 3 and ω 6 metabolism (Sordillo and Raphael, 2013) that they were investigated in this study. A further consideration is that acute phase proteins show differences in their response depending on the nature and severity of the inflammatory stimulus. Moreover, the type of acute phase protein with a greater diagnostic accuracy is also related to the animal species (Tóthová et al., 2011; Trevisi et al., 2011). Considering that our study involves buffaloes, the use of the most sensitive acute phase proteins identified in cattle (Hp and SAA; Tóthová et al., 2011, Trevisi et al., 2011) may not have been able to identify variations. This consideration should be further coupled with the cross-sectional experimental design that again would not allow us to assess changes over time further improving the assessment of the inflammatory status of the animals. The last consideration concerns the most sensitive acute phase proteins in cattle (Hp and SAA) that would seem to be associated with diseases not considered in this study such as mastitis, pneumonia, enteritis, peritonitis, endocarditis, abscesses, and endometritis (Tóthová et al., 2011; Trevisi et al., 2011).

  1. On line 84, the absence of a regular monitoring program for metabolic diseases, did not interfere with the results?

AU: Thank you for your questions. The absence of a monitoring program for metabolic diseases as a selection criterion was applied to increase the probability of identifying an adequate number of animals with metabolic alterations. In fact, the incidence of metabolic diseases in buffalo tends to be lower than in cattle (Bertoni et al., 1994; Purohit et al., 2013; Fiore et al. 2017). This inclusion criterion does not affect our results because the animals were submitted to a full clinical examination before being recruited in order to exclude the co-existence of other diseases.

  1. Which were the BCS used in the groups at the beginning of the experiment? How did it vary within the groups from the beginning of the experiment until the end? Lines 96 – 100

AU: Thank you for your questions. The current study aimed to assess the metabolic changes associated with different levels of energy deficit in Mediterranean Buffaloes at the early lactation stage. For the purposes of the manuscript, an observational, cross-sectional experimental design was used as reported in lines 112-113.  The body condition score was performed only at the beginning of the study (at sampling time) according to Guccione et al. (2016). Results from the two groups are reported in detail in Table 1, however, no additional scoring was performed because they did not fit the study design or its purposes.

  1. In lines 113 – 115, are the plasma/serum amounts for the pyrogallol amount, correct? “For each plasma sample, 250 µL of plasma was immediately transferred to Eppendorf tubes (1mL of serum/tube) containing 5 mg of pyrogallol to reduce FA oxidation “;

AU: Thank you for your questions. The protocol applied is an internal laboratory protocol based on Homemade methods drafted with the consultation of medical groups (Carnielli, V.P. et al., 1996a, 1996b; Cogo, P.E. et al., 1997, 1999). However, the amount of pyrogallol per sample is higher than that used in other articles (Eder, 1999; Puangkaew et al., 2004) in which 1 mg or less was used for 0.1 mL (100 µL). In our study, however, 5 mg was used for 0.25 mL (250 µL). In addition, studies conducted not on plasma but on erythrocyte membranes serum saline containing 5 g/L pyrogallol and 1 mmol/L EDTA to prevent oxidation (Vilaseca et al., 2010).

  1. Explain in line 114 the statement (1mL of serum/tube); was 250 μL of plasma added to a microtube that already contained 1 mL of serum?

AU: Thank you for your questions. We apologize it is a mistake in writing. We wanted to specify that the Eppendorf may contain up to 1 mL. In the case of plasma, 0.25 mL was placed with the addition of pyrogallol, while the entire Eppendorf (1 mL) was used in the case of serum.

Lines 116-120, we changed “For each plasma sample, 250 µL of plasma was immediately transferred to Eppendorf tubes (1mL of serum/tube) containing 5 mg of pyrogallol to reduce FA oxidation [15]. The samples were stirred until the pyrogallol was completely dis-solved. Also, serum samples were transferred to Eppendorf tubes (1mL of serum/tube) without pyrogallol.”

In

“For each plasma sample, 250 µL of plasma was immediately transferred to Eppendorf tubes (maximum capacity 1mL/tube) containing 5 mg of pyrogallol to reduce FA oxidation [15]. The samples were stirred until the pyrogallol was completely dis-solved. Also, serum samples were transferred to Eppendorf tubes (1mL of serum/tube) without pyrogallol.”

  1. The objectives emphasized in three moments, (lines 67 – 70; lines 24-26; lines 259-260) speak to energy levels and metabolic changes; How were energy levels determined? It is not described in the text. What metabolic changes have been identified? Very broad statement.

AU: Thank you for your questions. The exact quantification of the energy level was not calculated in this study. However, ketone bodies, especially BHB, are recognized as marker of the animal's energy balance. In fact, increased levels of BHB are associated to different degrees of NEB and therefore of energy imbalance in the animal (Sordillo and Raphael, 2013; Zhang and Ametaj, 2020; Horst et al., 2021). Our study used BHB as an indirect measure of two different levels of energy status of the animal. Certainly, it is also necessary to consider that there is no specific threshold value for buffalo for the identification of hyperketonemia, a metabolic disease due to NEB. Therefore, the mean BHB value of 0.4 mmol/L plus 3 standard deviations of 0.1 described in previous work of healthy buffaloes around 30 DIM (Fiore et al., 2017) was used to establish a subjective BHB cut-off of 0.7 mmol/ L as also explained in the study of Fiore et al. (2023).

Using the BHB for group classification, metabolic changes were assessed. Indeed, as explained in comment 1 by reviewer 2, the presence of an alteration of the lipid profile in itself represents an alteration of the animal metabolism. Furthermore, we attempted to understand the biological role(s) by fatty acid and/or category in order to improve the classification of these alterations.

  1. In the Abstract the results are not mentioned as proposed in the objectives, the metabolic alterations associated with different levels of energy deficit, only that there was a change in lipid classes and suggests a relationship in lipid metabolism and inflammation.

AU: Thank you for your questions. The limited discussion of the results was unfortunately due to a limited number of words (200) which in studies like this one, where the results are different, is particularly limiting. However, we have tried to modify the abstract as follows in order to improve the discussion of our results.

Lines 37-40, we changed “A total of 40 plasma FA was assessed in each lipid class. Six PL, 7 FFA, 6 TG, and 5 CE showed a significant difference and 2 PL, 3 FFA, and 1 TG were tended to significance. The changes among lipid classes suggest possible relationships with lipid metabolism and inflammation.”

In

“A total of 40 plasma FA was assessed in each lipid class. Amon FA, 8 PL, 7 FFA, 4 TG, and 4 CE increased according to BHB level while 3 FFA, 3 TG, and 1 CE decreased. The changes among lipid classes profiles suggested influence of inflammatory response, liver metabolism, and state of body lipid reserves.”

  1. It is important to note that we did not see in the paper how different energy deficits were identified in the animals; How to recognize that animals are at different levels of energy deficit?

AU: Thank you for your questions. As described in the comment 6 of reviewer 2, the exact quantification of energy deficit was not measured. Anyway, the BHB, the main ketone body, is a marker of energy deficit in animals. In fact, increased level of BHB indicates the presence of metabolic disease, also subclinical, due to an energy imbalance and consequently a deficit in animals (Sordillo and Raphael, 2013; Zhang and Ametaj, 2020; Horst et al., 2021). As done in our previous study (Fiore et al., 2023), the discriminating element for the recruitment of subjects was the measurement of BHB. Unfortunately, the negative energy balance in buffalo is still the reason for several investigations aiming to define the exact time in which it takes place, as well as the direct and indirect effects on metabolism and milk yield. The absence of a specific cut-off in buffaloes both for BHB (in case of ketosis) and for plasma lipid fractions related to ketone bodies proves the assumption. Based on the previous statements and the poor literature present, the authors decided to (i) use the BHB to identify subjects predisposed to an energy deficit as defined by Fiore et al., 2018 and (ii) refer to animals with “different energy levels” and/or “at risk of ketosis” along the manuscript. The exact assessment of the energy deficit represents an ambitious goal at which the authors are working to improve the health and welfare of the dairy buffalo.

  1. In lines 86 – 92, were the animals of the different groups fed the same amount?

AU: Thank you for your questions. Yes, the animals were fed the same amount because they belong to the same group. As reported in the manuscript n° 63 buffaloes were selected from a single dairy farm placed in Caserta (Campania Region, Italy) (lines 74-76).  All the animals within 50 days in milk (DIM) were randomly selected within the entire group of fresh buffaloes and milked twice a day (lines 78-80) and they were fed in the same manner. The line to which the Reviewer is referring is the inclusion criteria to select the farm were between a group of 10 regularly requesting consultancy services at the Veterinary Teaching Hospital – Didactic Mobile Clinic Service of the Department of Veterinary Medicine and Animal Production of Naples (Italy).

  1. In line 89, how much sodium bicarbonate?

AU: Thank you for your questions. The information has been added to the manuscript. We added in lines 94-95, “sodium bicarbonate (0.8% of the diet dry matter)”.

  1. In the conclusions (lines 367 – 373), it is stated that there are possible similarities in buffaloes at risk of hyper ketonemia with ketotic cows and suggests further investigation of ketosis in buffaloes because manifestation seems to differ between the two species; Unclear.

AU: Thank you for your questions. As indicated in lines 288-295 and 315-317, the changes found in MUFA levels in the present study seem to indicate a reduced gene expression of SCD1. This gene is downregulated in ketotic cows compared to hyperketonemic and healthy animals. The downregulation of this gene in buffaloes with BHB levels above 0.7, lower than the cut-off in cows, could represent a different metabolic condition between the two species. Obviously, this statement represents a hypothesis, as indicated in lines 293-295 and 317-320, and exact quantification of SCD1 gene expression in healthy and BHB-elevated buffalo should therefore be performed.

Round 2

Reviewer 2 Report

Dear Sirs.  Thank you for answering the questions.  After reading them and considering the text with the changes, we are in favor of its publication.